# ADAPTIVE MARGINAL SENSITIVITY WITH LIMITED RCT DATA FOR CATE ESTIMATION

## ABSTRACT

The conditional average treatment effect (CATE) is pivotal for personalized decision-making across numerous domains. While observational studies (OBS) are a primary data source for estimating CATE, they are susceptible to bias from unmeasured confounding. The marginal sensitivity model (MSM) addresses this by quantifying the robustness of causal conclusions to such confounding via a sensitivity parameter, $\Gamma$. However, a significant limitation of MSM is the need for researchers to subjectively specify $\Gamma$, which lacks a data-driven basis and undermines the reliability of inferences. Recent methods that use randomized controlled trial (RCT) data to calibrate $\Gamma$ are promising but critically depend on having a large RCT sample, which is often unavailable in practice. To overcome this limitation, we propose the Bayesian Marginal Sensitivity Calibration (BMSC) framework. BMSC learns the sensitivity parameter $\Gamma$ directly from fused RCT and OBS data, shifting the paradigm from subjective specification to data-driven estimation. Our approach constructs a CATE envelope from OBS, calibrates $\Gamma$ by assessing the alignment with RCT estimates, and produces robust CATE intervals with valid coverage guarantees. Theoretical analysis and extensive experiments show that BMSC provides sharper, more accurate intervals than methods using subjective $\Gamma$ values, and remains effective even when the RCT sample size is very small. This work provides a practical and robust solution for sensitivity analysis in real-world settings with limited experimental data.

## 1 INTRODUCTION

Personalized decision-making increasingly relies on accurate estimation of the conditional average treatment effect (CATE) across fields such as medicine (Collins & Varmus, 2015; Kent et al., 2010; 2018), economics (Kitagawa & Tetenov, 2018; Athey & Imbens, 2016; Knaus et al., 2021), and marketing (Radcliffe & Surry, 2011; Ascarza, 2018; Hitsch et al., 2024). Reliable CATE estimation is thus essential for implementing precise interventions at scale. In practice, observational studies (OBS) serve as a key data source for estimating CATE due to their cost-effectiveness and ease of data collection (Oprescu & Kallus, 2024; Gilmartin-Thomas et al., 2018). However, a major limitation of OBS-based CATE estimation is the potential presence of **unmeasured confounding**, which can bias causal conclusions (Gilmartin-Thomas et al., 2018).

The *marginal sensitivity model (MSM* provides a sensitivity analysis framework for assessing the impact of unmeasured confounding in observational studies by introducing a sensitivity parameter $\Gamma$ that bounds the possible deviation between the true propensity score and an estimated nominal model. This approach quantifies the strength of unmeasured confounding required to alter causal conclusions, offering localized, interpretable bias control applicable to both population-level average treatment effects and individual- or subgroup-level CATEs for partial identification and interval estimation (Rosenbaum, 2002; Imbens, 2003; Kallus & Zhou, 2018; Kallus et al., 2019). The MSM framework offers the distinct advantage of not requiring instrumental variables, relying instead on observed data and a sensitivity parameter $\Gamma$ to assess unmeasured confounding. This makes it broadly applicable in settings where valid instruments are unavailable.

However, A central limitation of the MSM framework is that **the sensitivity parameter** $\Gamma$ must be specified by the researcher in advance rather than being determined empirically from the data. This requirement introduces subjectivity into the analysis, as there is no universal scale or statistically

testable basis for choosing an appropriate $\Gamma$ value. As a result, conclusions can vary substantially across studies depending on the analyst's subjective judgment, undermining cross-study comparability and reproducibility. Moreover, causal inferences may become sensitive to the chosen $\Gamma$, producing overly narrow intervals that fail to maintain nominal coverage when $\Gamma$ is set too small, or yielding wide and uninformative intervals when $\Gamma$ is overly conservative. Such ambiguity can critically affect decision-making, potentially leading to reversed or misleading policy recommendations (Keele, 2015; Ding & VanderWeele, 2016; Cinelli & Hazlett, 2020). This reliance on pre-specification represents a major practical barrier to the reliable use of sensitivity analysis in real-world settings. Recent methods estimate $\Gamma$ from RCT data but typically need large trials, often infeasible given data scarcity and high costs (De Bartolomeis et al., 2024). This reliance on subjective pre-specification or abundant RCTs hinders practical deployment of reliable sensitivity analysis.

To address this open challenge, particularly the critical limitation of limited RCT data, we introduce **Bayesian Marginal Sensitivity Calibration (BMSC)**, a framework that estimates the sensitivity parameter $\Gamma$ from fused data and leverages it to guide CATE estimation. This framework accomplishes this through a three-stage method: it first constructs a CATE envelope from observational data via sensitivity analysis, then calibrates $\Gamma$ by evaluating how well the RCT estimates fit within this envelope, and finally uses the estimated $\Gamma$ to produce robust CATE intervals. This approach represents a paradigm shift from testing pre-specified values of $\Gamma$ to actively learning a data-driven sensitivity parameter, thereby eliminating the dependence on large-scale RCTs for reliable sensitivity calibration. Theoretically, we establish that the proposed estimator provides valid coverage guarantees while achieving sharper interval widths compared to methods relying on subjective, often conservative $\Gamma$ choices. Extensive experiments demonstrate that BMSC delivers reliable and informative inferences even when the RCT sample size is too small for existing calibration methods to be effective, thus directly addressing the practical challenge of limited RCT data in sensitivity analysis. The main contributions of this work are:

- We propose a Bayesian marginal sensitivity calibration (BMSC) that learns $\Gamma$ from fused data, propagates its posterior to CATE inference, and yields tighter, coverage-guaranteed intervals without subjective $\Gamma$ choices.

- To our knowledge, this is the first framework that relaxes the heavy RCT sample size requirement for estimating $\Gamma$, thereby providing a practical solution for stable sensitivity calibration even with limited trial data.

- On synthetic and real data, BMSC attains more accurate CATE estimates and better RCT sample efficiency than threshold-testing MSM calibrations and alignment-centric baselines, especially when RCT samples are scarce.

## 2 RELATED WORKS

### 2.1 MARGINAL SENSITIVITY MODELS

The Marginal Sensitivity Model formalizes ignorability violations via a scalar parameter that bounds the influence of unmeasured confounding, producing interpretable CATE bounds when the parameter is fixed (Tan, 2006; Rosenbaum, 2002). This parameter is typically chosen using domain knowledge or external calibration (Hsu & Small, 2013; Rosenbaum, 2004), and most estimation methods, including model-assisted approaches (Tan, 2024), condition on a fixed value. While sharp bounds have been derived under MSM constraints (Zhao et al., 2019; Frauen et al., 2023), data-driven estimation of the sensitivity parameter from fused RCT–OBS evidence remains uncommon: current trial-anchored methods often only yield lower bounds, require large samples, and perform poorly under weak confounding (De Bartolomeis et al., 2024). Thus, MSM **has not been fully integrated into fusion estimators** as a learnable component. To migrate this, we propose to identify the sensitivity level directly from fused data, and propagates this sensitivity to produce valid CATE intervals even in small RCT samples.

### 2.2 QUANTIFYING UNMEASURED CONFOUNDING THROUGH FUSING DATA

Incorporating measures of residual unmeasured confounding into data fusion reframes robustness as a constraint that restricts plausible CATE values and regulates OBS influence. Existing approaches

fall into three categories: (i) using trials to bound hidden bias, though these often yield only lower bounds with limited power under weak confounding (De Bartolomeis et al., 2024); (ii) sensitivity-aware fusion, which embeds bias parameters to narrow effect estimates or shift them toward RCT results (Jiang et al., 2024; Yu et al., 2024); (iii) trial-referenced benchmarking to pre-evaluate OBS reliability before fusion (Dahabreh et al., 2022; Forbes & Dahabreh, 2020; Matthews et al., 2022; Thorlund et al., 2024). While improving transparency, these methods **do not treat confounding strength as an estimable parameter** that propagates uncertainty into individual CATE estimates. A unified framework is needed to quantify unmeasured confounding from fused data and propagate it to CATE uncertainty in a single model, enabling joint robustness–efficiency optimization, particularly in small trials or under weak confounding.

### 2.3 DATA FUSION FOR TREATMENT EFFECT ESTIMATION

Data fusion methods for heterogeneous treatment effects fuse RCT and OBS data in three primary ways: (i) Extrapolation and generalizability weighting, which reweights the RCT sample to match a target OBS population, enabling effect transportability (Pearl & Bareinboim, 2011; Bareinboim & Pearl, 2016; Cole & Stuart, 2010; Dahabreh et al., 2024; Stuart et al., 2018; Dahabreh et al., 2020; Degtiar & Rose, 2023); (ii) Representation or meta-learning two-stage fusion, where effect heterogeneity is learned from OBS and refined via RCT using X-/R-learners, including pretraining on biased data followed by calibration with RCT (Hatt et al., 2022; Künzel et al., 2019; Li et al., 2023a;b); (iii) Controlled borrowing and robust fusion, which jointly model RCT and OBS data with regularization via partial pooling, power priors, Gaussian processes, or adaptive weighting, often incorporate instrumental variables (Dimitriou et al., 2024; Ibrahim & Chen, 2000; Cheng & Cai, 2021; Asiaee et al., 2023; Oprescu & Kallus, 2024; Lin et al., 2025). Recent surveys synthesize identification and estimation strategies (Colnet et al., 2024; Degtiar & Rose, 2023; Shi et al., 2023). However, most methods focus on efficiency or external validity without formal robustness quantification, which are commonly **lack of in-model estimation of unmeasured confounding** to guide data fusion. Our approach addresses this by learning a sensitivity parameter directly from fused RCT–OBS data and propagating it to MSM-based intervals, embedding robustness within the fusion process itself.

## 3 PRELIMINARIES AND PROBLEM SETUP

We consider the problem of fusing two independent data sources:

- An observational study, denoted $\mathcal{D}_{\text{obs}} = \{X_i, T_i, Y_i\}_{i=1}^{N_{\text{obs}}}$, in which treatment assignment $T$ may depend on both observed covariates $X$ and unobserved confounders $U$.
- A randomized controlled trial, denoted $\mathcal{D}_{\text{rct}} = \{X_i, T_i, Y_i\}_{i=1}^{N_{\text{rct}}}$, where by design the treatment is randomized and hence satisfies $(Y(1), Y(0)) \perp T$.

Our goal is to estimate CATE, defined as $\tau(x) = \mathbb{E}[Y(1) - Y(0) \mid X = x]$, in a way that leverages the complementary strengths of both datasets. A key challenge is to explicitly quantify and adjust for bias arising from unmeasured confounding $U$ in the observational data.

**Marginal Sensitivity Models.** MSMs address unmeasured confounding by introducing a parameter $\Gamma \geq 1$ that bounds the deviation between the true and estimated propensity scores. This induces a set of $\Gamma$-dependent bounds on the counterfactual means and, consequently, on the CATE, framing confounding bias as a constrained estimation problem.

**Current Data Fusion Methods.** Existing approaches combine the internal validity of the RCT with the broad coverage of the observational data. They typically transport trial estimates, leverage effect heterogeneity from the observational study, or perform controlled data pooling. Collectively, they aim to enhance precision and generalizability by using the observational data for efficiency while relying on the RCT for identification.

**Gap in Current Methods.** Existing approaches suffer from:

- Classical MSM treats $\Gamma$ as an exogenously chosen sensitivity parameter rather than learning it from data, and does not propagate its uncertainty to CATE, which can yield overly wide or miscalibrated intervals.

- Most data fusion methods lack in-model quantification of unmeasured confounding, relying instead on post-hoc sensitivity analyses; when covariate overlap is limited or variable definitions are not harmonized, they are prone to negative transfer and bias amplification.

**Problem Statement.** We aim to develop a unified framework that: (i) quantifies the sensitivity parameter $\Gamma$ from limited RCT plus OBS, (ii) propagates the uncertainty in $\Gamma$ to the CATE estimates.

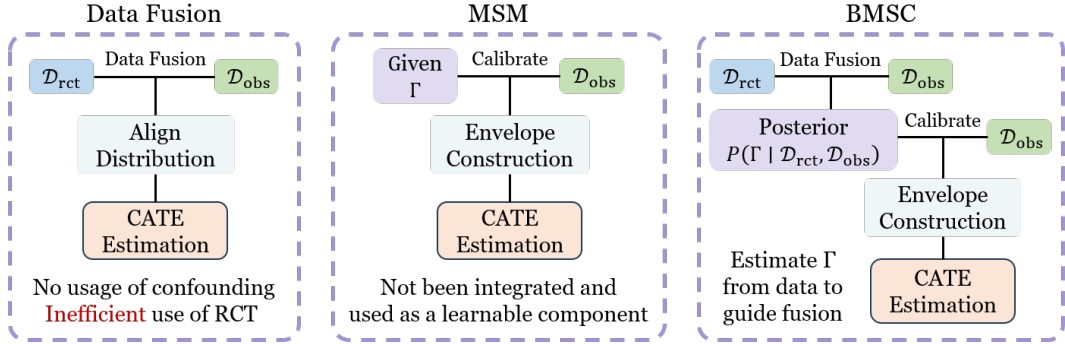

Figure 1: Problem Setup: Data Fusion, MSM, and BMSC.

## 4 BAYESIAN MARGINAL SENSITIVITY CALIBRATION

In this section, we introduce *Bayesian Marginal Sensitivity Calibration (BMSC)*. Our framework is designed to learn the strength of unmeasured confounding ($\Gamma$) from fused data, and then use this estimate to produce robust, uncertainty-aware CATE intervals. Unlike methods that require full distributional alignment, BMSC uses a sensitivity model to translate $\Gamma$ into identifiable CATE bounds, creating a fusion bridge that is calibrated through a Bayesian pipeline.

### 4.1 SENSITIVITY MODEL AS A FUSION BRIDGE

The foundation of our approach is a sensitivity model that quantifies the impact of unmeasured confounders $U$ by constraining how much they can influence treatment assignment. We begin by formally stating the model.

**Definition 1 (Marginal Sensitivity Model (MSM))** *(Tan, 2006; Zhao et al., 2019) Let $e_t(x) = \mathbb{P}(T = t \mid X = x)$ be the nominal propensity and $e_t(x, y) = \mathbb{P}(T = t \mid X = x, Y(t) = y)$ the complete propensity. The MSM posits a global confounding strength $\Gamma \geq 1$ such that, for any $t \in \{0, 1\}$, $x \in \mathcal{X}$, $y \in \mathcal{Y}$,*

$$\frac{1}{\Gamma} \leq \frac{(1 - e_t(x))\, e_t(x, y)}{e_t(x)\, (1 - e_t(x, y))} \leq \Gamma. \tag{1}$$

This condition bounds the degree to which unmeasured confounders affect treatment assignment odds. For a given $\Gamma$, this constraint implies that the true CATE $\tau(x)$ must lie within an identifiable interval. This is formalized below.

**Theorem 1 (CATE Bounds under MSM)** *For any $\Gamma \geq 1$, the MSM implies that the true CATE $\tau(x)$ lies within an identifiable interval:*

$$\begin{aligned}
\tau(x) &\in [L(x, \Gamma),\, U(x, \Gamma)] \\
&= [\omega(x) + \underline{\eta_1}(x, \Gamma) - \overline{\eta_0}(x, \Gamma),\, \omega(x) + \overline{\eta_1}(x, \Gamma) - \underline{\eta_0}(x, \Gamma)],
\end{aligned} \tag{2}$$

*where the components are defined as:*

$$\omega(x) = \mu_1(x) - \mu_0(x) = E(Y \mid T = 1, X = x) - E(Y \mid T = 0, X = x),$$

$$\underline{\eta_t}(x, \Gamma) = (\frac{1}{\Gamma} - 1)(1 - e_t(x))\mu_t(x), \quad \overline{\eta_t}(x, \Gamma) = (\Gamma - 1)(1 - e_t(x))\mu_t(x).$$

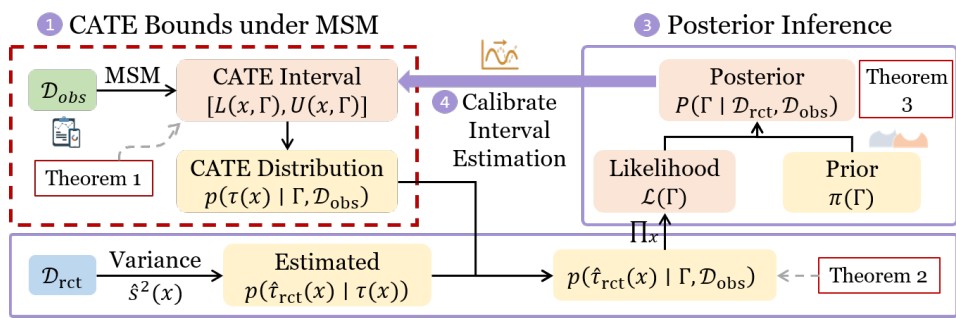

Figure 2: Workflow for Proposed Method (BMSC).

*Here, $\omega(x)$ is the naive CATE estimate from the observational data, and $\eta_t$, $\overline{\eta_t}$ are bias correction terms that depend on the strength of unmeasured confounding $\Gamma$. Derivations and sharpness proofs are provided in Appendices A.1.2 and A.1.3.*

These bounds form a fusion bridge: the observational data determine a feasible envelope $[L(x,\Gamma), U(x,\Gamma)]$ for the CATE, while the RCT provides identified but noisy targets that must lie within this envelope if the true confounding strength is $\Gamma$.

### 4.2 BAYESIAN CALIBRATION OF THE CONFOUNDING STRENGTH $\Gamma$

We now describe how to calibrate $\Gamma$ using the RCT data. The core idea is to find the value of $\Gamma$ that makes the observational bounds most compatible with the RCT estimates. We adopt a Bayesian approach, combining a prior with a likelihood derived from the RCT.

#### 4.2.1 PRIOR ON $\Gamma$.

We specify a weakly informative prior $\pi(\Gamma)$ on $[1, \Gamma_{\max}]$, treating $\Gamma$ as an unknown parameter to be estimated. In practice, the prior may be centered and scaled using empirical benchmarks, such as observed covariate imbalances (Rosenbaum, 2002; Hsu & Small, 2013).

#### 4.2.2 LIKELIHOOD FROM THE TRIAL.

For a fixed $\Gamma$, the model restricts the true CATE $\tau(x)$ to the interval $[L(x,\Gamma), U(x,\Gamma)]$. The likelihood measures how plausible the observed RCT estimate $\hat{\tau}_{\mathrm{rct}}(x)$ is, given that the true effect lies within this bounds. Taking the RCT-based estimator as an unbiased noisy measurement of $\tau(x)$ with variance $\hat{s}^2(x)$, we integrate over the constrained support to obtain a marginal likelihood for each covariate stratum (assumptions and proof in Appendix A.1.4).

**Theorem 2 (Marginal Likelihood for Data Fusion)** *The marginal likelihood of observing the RCT estimate $\hat{\tau}_{\mathrm{rct}}(x)$ given confounding strength $\Gamma$ has the closed-form expression:*

$$p\left(\hat{\tau}_{\mathrm{rct}}(x) \mid \Gamma, \mathcal{D}_{\mathrm{obs}}\right) = \frac{\Phi\left(\frac{U(x,\Gamma)-\hat{\tau}_{\mathrm{rct}}(x)}{\hat{s}(x)}\right) - \Phi\left(\frac{L(x,\Gamma)-\hat{\tau}_{\mathrm{rct}}(x)}{\hat{s}(x)}\right)}{U(x,\Gamma) - L(x,\Gamma)}, \tag{3}$$

*where $\Phi(\cdot)$ denotes the standard normal cumulative distribution function.*

The overall likelihood is the product over all strata: $\mathcal{L}(\Gamma) = \prod_x p\left(\hat{\tau}_{\mathrm{rct}}(x) \mid \Gamma, \mathcal{D}_{\mathrm{obs}}\right)$. This function statistically measures how well a candidate $\Gamma$ aligns the RCT estimates with the predicted envelope derived from the observational data.

#### 4.2.3 POSTERIOR FOR $\Gamma$.

Combining the likelihood $\mathcal{L}(\Gamma)$ with the prior $\pi(\Gamma)$, we obtain the posterior distribution over the confounding strength:

---

**Algorithm 1** BMSC

---

**Input:** $\mathcal{D}_{\text{obs}}, \mathcal{D}_{\text{rct}}$, prior $\pi(\Gamma)$, grid $\mathcal{G} \subseteq [1, \Gamma_{\max}]$
**Output:** Posterior $P(\Gamma \mid \mathcal{D}_{\text{obs}}, \mathcal{D}_{\text{rct}})$; CATE intervals
 1: **Observational:** For each $x$, estimate $e_t(x), \mu_t(x), \omega(x) = \mu_1(x) - \mu_0(x)$.
 2: **RCT:** For each $x$, compute $\hat{\tau}_{\text{rct}}(x)$ and $\hat{s}(x)$.
 3: **for** $\gamma \in \mathcal{G}$ **do**
 4:    **MSM bounds:** Compute $L(x, \gamma), U(x, \gamma)$ (Eq. equation 2).
 5:    **Marginal likelihood:** $\ell_x(\gamma)$ (Eq. equation 3).
 6:    **Joint likelihood:** $\mathcal{L}(\gamma) = \prod_x \ell_x(\gamma)$.
 7: **end for**
 8: **Posterior for $\Gamma$:** $P(\Gamma \mid \mathcal{D}_{\text{obs}}, \mathcal{D}_{\text{rct}}) \propto \mathcal{L}(\Gamma)\pi(\Gamma)$ (Eq. equation 4); normalize on $\mathcal{G}$.
 9: **Summaries:** Report posterior mean/median for $\Gamma$.
10: **CATE intervals:** Using $\hat{\Gamma}$ (mean/median), output $[L(x, \hat{\Gamma}), U(x, \hat{\Gamma})]$ for each $x$.

---

**Theorem 3 (Posterior Inference for Confounding Strength)** *Assuming conditional independence of the RCT estimation errors across covariate strata $x$ given $\Gamma$, the posterior distribution is:*

$$P(\Gamma \mid D_{\text{rct}}, D_{\text{obs}}) \propto \mathcal{L}(\Gamma) \cdot \pi(\Gamma) = \prod_x p\left(\hat{\tau}_{\text{rct}}(x) \mid \Gamma, \mathcal{D}_{\text{obs}}\right) \cdot \pi(\Gamma). \tag{4}$$

Since $\Gamma$ is one-dimensional, we evaluate equation 4 on a dense grid over $[1, \Gamma_{\max}]$ and normalize. This posterior quantitatively integrates trial evidence with the observational model, forming a unified, uncertainty-aware estimate of the confounding strength.

**Remark 1** *In summary, we specify a prior for $\Gamma$, derive a closed-form marginal likelihood by integrating the trial sampling model over the bounds implied by the observational data, and combine these components to obtain a tractable posterior. This posterior encapsulates uncertainty about hidden confounding.*

### 4.3 ESTIMATION FOR $\Gamma$ AND CATE

The following procedure outlines the estimation of both the confounding strength $\Gamma$ and the conditional average treatment effect $\tau(x)$.

**Posterior summary for $\Gamma$.** From the $S$ posterior samples $\{\Gamma^{(i)}\}_{i=1}^S$, we compute summary statistics including: the posterior mean $\hat{\Gamma}_{\text{mean}} = \text{mean}\{\Gamma^{(i)}\}_{i=1}^S$ and the posterior median: $\hat{\Gamma}_{\text{median}} = \text{median}\{\Gamma^{(i)}\}_{i=1}^S$.

**CATE interval estimation.** We take $\hat{\Gamma}$ (either the posterior mean or median) as the point estimate of the confounding strength, and obtain the corresponding CATE interval estimate:

$$\tau(x) \in \left[L(x, \hat{\Gamma}), U(x, \hat{\Gamma})\right],$$

which provides a confounding-adjusted interval estimate for the conditional average treatment effect that incorporates the estimated strength of unmeasured confounding.

**Remark 2** *This approach provides both a quantitative estimate of the confounding strength and a corresponding interval estimate for the CATE that explicitly accounts for the impact of unmeasured confounders. The Bayesian framework ensures proper propagation of uncertainty from the estimated $\Gamma$ to the final CATE intervals.*

### 4.4 COMPUTATIONAL COMPLEXITY ANALYSIS

The overall complexity of the proposed BMSC is derived as $O\left(N_{\text{obs}} + N_{\text{rct}} + GM + SM + G\log G\right)$, where $N_{\text{obs}}$ and $N_{\text{rct}}$ are the sample sizes of the observational and trial datasets, $M$ is the number of covariate strata used for calibration, $G = |\mathcal{G}|$ is the grid size for $\Gamma$, and $S$ is the number of posterior draws used to propagate uncertainty to CATE (detailed derivation provided in Appendix A.1.5).

## 5 EXPERIMENTS

**Benchmarks.** We evaluate the performance of BMSC in estimating both the confounding strength and the conditional average treatment effect (CATE) using synthetic datasets and a real-world dataset from the ACTG clinical trial (Hammer et al., 1996). Further details on the datasets are provided in Appendix A.2.1.

**Baselines.** We compare BMSC with state-of-the-art baselines from two areas:

(i) Marginal Sensitivity Models: fMSM (Kallus et al., 2019), PB-IPW (Zhao et al., 2019).

(ii) Quantifying Unmeasured Confounding through Fusing Data: QCLB (De Bartolomeis et al., 2024);

(iii) Data Fusion for Treatment Effect Estimation: SF, ST (Gu et al., 2023), Two-step (Kallus et al., 2018), and CORNets (Hatt et al., 2022). (See details of baselines in Appendix A.2.2).

**Evaluation Protocols.** We evaluate the proposed BMSC in a variety of scenarios, including:

(i) Marginal Sensitivity Models: Compare the miscoverage rate (proportion of intervals failing to cover the true CATE) and average interval width (reflecting estimation precision) of BMSC against traditional MSM approaches with $\Gamma$ set correctly, overly large, or overly small, in order to assess robustness and interval efficiency under realistic settings where the true $\Gamma$ is unknown.

(ii) Quantifying Unmeasured Confounding through Fusing Data: Compare the estimated confounding strength $\hat{\Gamma}$ (posterior mean for BMSC; lower bound for QCLB) across varying true confounding levels $\Gamma_{\text{true}}$ and RCT sample sizes $N_{\text{rct}}$;

(iii) Data Fusion for Treatment Effect Estimation: Evaluate the accuracy and calibration of CATE estimates by examining whether $95\%$ confidence intervals (for point estimators) and interval estimates cover the true effect $\tau(x)$, and compare their widths across covariate values $x$.

**Questions.** The empirical experiments are performed around the following three questions:

(i) How does BMSC perform relative to traditional MSM methods that rely on pre-specified $\Gamma$ values, particularly in terms of coverage control and interval width when $\Gamma$ is misspecified?

(ii) How accurately and with what sample efficiency does BMSC estimate the confounding strength $\Gamma$ relative to the QCLB baseline?

(iii) How do the accuracy and sample efficiency of BMSC's CATE estimates compare with other data-fusion baselines?

(iv) How do model parameters influence performance and sample efficiency?

### 5.1 COMPARISON WITH TRADITIONAL MSM APPROACHES

We compare BMSC against traditional MSM methods with $\Gamma$ set correctly, overly large, or overly small, using miscoverage rate and average interval width as evaluation metrics. The results demonstrate that BMSC achieves comparable performance to traditional MSM with the correct $\Gamma$: both maintain a miscoverage rate of 0 and exhibit similar average interval widths. In contrast, traditional MSM with an underspecified $\Gamma$ leads to substantial miscoverage and excessively narrow intervals, indicating undercoverage and inflated type I error. Conversely, traditional MSM with an overspecified $\Gamma$ yields valid coverage but produces overly conservative intervals with widths exceeding 20.79, resulting in inefficient inference. BMSC avoids these pitfalls by automatically learning a data-driven $\Gamma$ close to the true value, achieving both valid coverage and interval efficiency without requiring prior knowledge of the confounding strength.

### 5.2 EVALUATION OF CONFOUNDING STRENGTH ESTIMATION

We construct a controlled synthetic study with known ground truth, fixing $N_{\text{obs}}$ and varying the RCT size $N_{\text{rct}}$ and the true confounding strength $\Gamma_{\text{true}}$. For each setting, BMSC returns the posterior mean of $\Gamma$, while QCLB reports a lower bound; NA denotes that QCLB yields no informative bound at the nominal test level. Table 2 shows that BMSC closely recovers $\Gamma_{\text{true}}$ across all regimes and

Table 1: Comparison of BMSC with traditional MSM methods under different $\Gamma$ specifications

| Method | $\Gamma$ specification | Miscoverage rate | Avg. width |
|--------|------------------------|------------------|------------|
| BMSC | learned from data | 0.00 | 11.75 |
| fMSM | under ($\Gamma = 1.01$) | 0.38 | 0.16 |
| fMSM | true ($\Gamma = 2.0$) | 0.00 | 11.83 |
| fMSM | over ($\Gamma = 2.99$) | 0.00 | 20.95 |
| PB-IPW | under ($\Gamma = 1.01$) | 0.04 | 0.36 |
| PB-IPW | true ($\Gamma = 2.0$) | 0.00 | 12.13 |
| PB-IPW | over ($\Gamma = 2.99$) | 0.00 | 21.48 |

is only weakly sensitive to $N_{\text{rct}}$. QCLB performs poorly when $N_{\text{rct}}$ is small, failing to produce informative bounds at low or moderate confounding and returning only a conservative value at high confounding. Even with $N_{\text{rct}} = 5000$, QCLB remains conservative. Overall, BMSC is uniformly more informative and markedly more sample efficient for confounding quantification.

Table 2: Confounding strength estimation across RCT sample sizes. Each cell reports $|\hat{\Gamma} - \Gamma_{\text{true}}|$.

| $N_{\text{rct}}$ | Method | $\Gamma_{\text{true}} = 1.5$ | $\Gamma_{\text{true}} = 3.0$ | $\Gamma_{\text{true}} = 6.0$ |
|------------------|--------|------------------------------|------------------------------|------------------------------|
| 200 | QCLB | NA | NA | 3.000 |
| | BMSC | 0.107 | 0.001 | 0.076 |
| 5000 | QCLB | 0.200 | 0.500 | 0.500 |
| | BMSC | 0.053 | 0.000 | 0.076 |

## 5.3 ASSESSMENT OF TREATMENT EFFECT ESTIMATION

We compare BMSC against data-fusion baselines on both synthetic and ACTG benchmarks, using mean absolute error (MAE) of CATE across various randomized-trial budgets $N_{\text{rct}}$. On both datasets, BMSC consistently and robustly achieves the lowest MAE for a given $N_{\text{rct}}$, with the greatest improvements observed at small to moderate trial sizes, demonstrating its superior sample efficiency. The baselines generally exhibit substantially higher error at low $N_{\text{rct}}$, which decreases only gradually as the trial size increases. Among these, Two-step consistently trails BMSC and requires substantially larger trials to match its accuracy; CORNets consistently underperforms. The SF baseline improves slowly with larger $N_{\text{rct}}$ but remains less accurate than BMSC, while ST shows unstable performance, occasionally competitive at specific budgets but not robust overall. BMSC achieves target accuracy with far fewer randomized samples than alternatives and delivers uniformly better CATE estimation on both benchmarks.

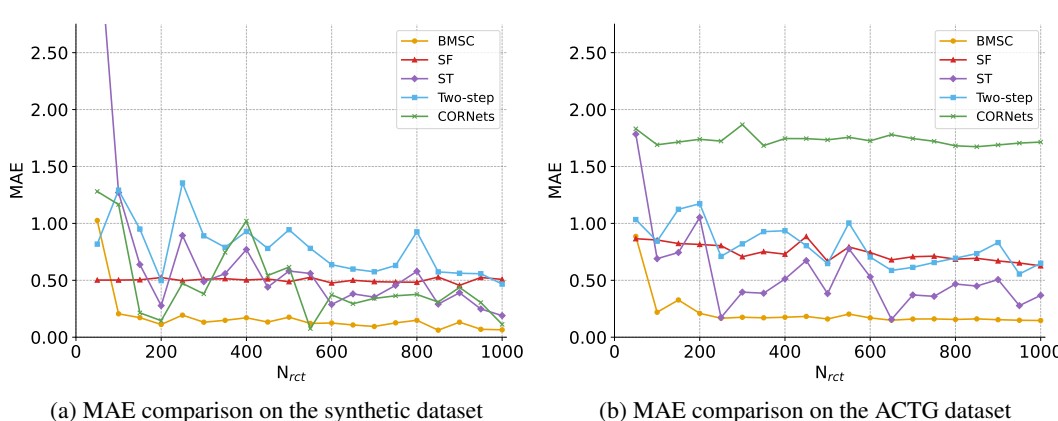

(a) MAE comparison on the synthetic dataset     (b) MAE comparison on the ACTG dataset

Figure 3: Comparison of CATE estimation performance across methods.

## 5.4 SENSITIVITY ANALYSIS OF MODEL PARAMETERS

We conduct a sensitivity analysis to assess how the RCT sample size $N_{\text{rct}}$, the prior dispersion on the confounding strength $\Gamma$ (log scale $\sigma_{\text{prior}}$), and the choice of posterior point summary (mean vs. median) affect the accuracy of BMSC. Table 3 and Figure 4 show three consistent patterns. First, increasing $N_{\text{rct}}$ generally improves accuracy across settings; gains can taper when confounding is very strong, but the overall trend is monotone. Second, more diffuse priors degrade accuracy due to posterior skew, whereas mildly informative priors yield the best bias–variance trade-off. Third, the median is typically more robust under skewed posteriors or small $N_{\text{rct}}$, while the mean can be slightly more efficient when the posterior is approximately symmetric or $N_{\text{rct}}$ is large.

Table 3: Sensitivity of confounding-strength estimation to RCT size, prior dispersion, and posterior summary. Each cell reports $|\hat{\Gamma} - \Gamma_{\text{true}}|$ as "mean / median".

| $\Gamma_{\text{true}}$ | $N_{\text{rct}}$ | $\sigma_{\text{prior}} = 0.25$ | $\sigma_{\text{prior}} = 0.50$ | $\sigma_{\text{prior}} = 1.00$ |
|---|---|---|---|---|
| 1.5 | 200 | 0.107 / 0.038 | 0.428 / 0.212 | 1.128 / 0.560 |
| | 5000 | 0.053 / 0.010 | 0.324 / 0.112 | 0.970 / 0.406 |
| 3.0 | 200 | 0.001 / 0.094 | 0.074 / 0.276 | 0.275 / 0.384 |
| | 5000 | 0.000 / 0.095 | 0.030 / 0.313 | 0.135 / 0.529 |
| 6.0 | 200 | 0.076 / 0.214 | 0.784 / 1.031 | 1.902 / 2.480 |
| | 5000 | 0.076 / 0.214 | 0.788 / 1.033 | 2.003 / 2.589 |

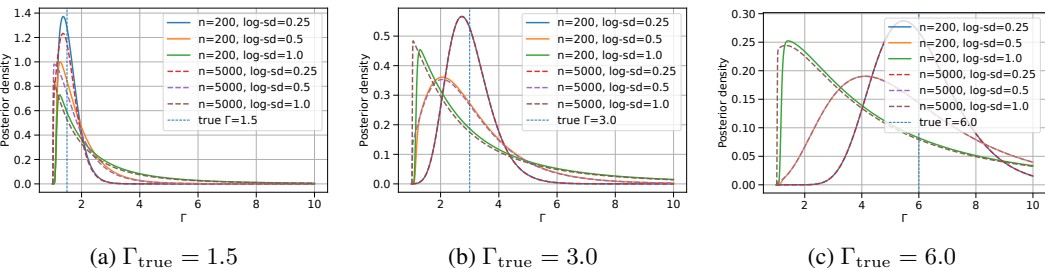

(a) $\Gamma_{\text{true}} = 1.5$        (b) $\Gamma_{\text{true}} = 3.0$        (c) $\Gamma_{\text{true}} = 6.0$

Figure 4: Posterior over the sensitivity parameter $\Gamma$.

## 6 CONCLUSION

We addressed the fundamental critical challenge of subjective specification of the sensitivity parameter $\Gamma$ in marginal sensitivity models. While recent methods use randomized controlled trial (RCT) data to calibrate $\Gamma$, they require large trial samples that are often unavailable in practice. We introduce the novel Bayesian Marginal Sensitivity Calibration (BMSC), a framework that learns $\Gamma$ directly from fused observational and RCT data. BMSC constructs a sensitivity-based CATE envelope from observational data, calibrates $\Gamma$ using trial evidence, and effectively produces robust CATE intervals with valid coverage guarantees.

Theoretically, BMSC establishes a tractable link between RCT estimates and the observational sensitivity model, turning $\Gamma$ from a fixed parameter into a data-driven quantity. Empirically, BMSC achieves tighter, better-calibrated intervals than methods using subjective $\Gamma$ values and remains effective with minimal RCT data, where conventional calibration fails. By enabling reliable sensitivity analysis with limited trial data, BMSC provides a practical solution for credible CATE estimation in real-world applications. Across synthetic and real-world experiments, BMSC consistently achieves superior calibration and precision in CATE interval estimation relative to state-of-the-art alternatives. Its robust performance in data-scarce scenarios underscores the practical value of propagating $\Gamma$ uncertainty for credible causal inference.

## ETHICS STATEMENT

This work does not raise any ethical concerns. All experiments are conducted on publicly available datasets, and no human subjects or sensitive attributes are involved. We confirm that our study complies with the ICLR Code of Ethics.

## REPRODUCIBILITY STATEMENT

We provide detailed descriptions of our framework, theoretical results, and experimental settings in the paper and appendix. All datasets used are publicly available, and the current description of our method is sufficient for full reproducibility. If the paper is accepted, we will be glad to release the complete implementation to further support the research community.

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

## A  APPENDIX

### A.1  THEORETICAL ANALYSIS

#### A.1.1  THEORETICAL ANALYSIS OF SAMPLE EFFICIENCY

Fix a covariate value $x$ and a desired CATE interval half–width $\varepsilon$ with coverage at least $1-\alpha$. Under BMSC, the RCT is used only to calibrate the scalar confounding strength $\Gamma$, and the resulting CATE interval width at $x$ equals $W_x(\Gamma) = U(x,\Gamma) - L(x,\Gamma)$. If the posterior standard error of $\hat{\Gamma}$ scales as $\mathrm{SE}(\hat{\Gamma}) = \kappa/\sqrt{N_{\mathrm{rct}}}$, a first–order delta method yields the incremental half–width due to estimating $\Gamma$,

$$\Delta_x \approx z_{1-\alpha/2} \left| \partial_\Gamma W_x(\Gamma^\star) \right| \mathrm{SE}(\hat{\Gamma}) = z_{1-\alpha/2} \left| \partial_\Gamma W_x(\Gamma^\star) \right| \frac{\kappa}{\sqrt{N_{\mathrm{rct}}}}.$$

Requiring $\Delta_x \leq \varepsilon$ gives the parametric-rate requirement

$$N_{\mathrm{rct}}^{\mathrm{Cal}} \gtrsim \frac{z_{1-\alpha/2}^2 \, \kappa^2 \left( \partial_\Gamma W_x(\Gamma^\star) \right)^2}{\varepsilon^2} = O\!\left( \varepsilon^{-2} \right).$$

By contrast, Two-step debiasing fits a *function* $\delta(x)$ using RCT data, and the RCT sample size must control the estimation error of this function class. For common classes (omitting constants and benign logarithmic factors), the required $N_{\mathrm{rct}}$ to achieve the same half–width $\varepsilon$ satisfies

$$N_{\mathrm{rct}}^{\mathrm{Two\text{-}step}} \gtrsim \begin{cases} s \log p \, \varepsilon^{-2}, & \text{(sparse linear classes),} \\ \varepsilon^{-2d/r}, & \text{(nonparametric classes in } d \text{ dimensions with smoothness } r\text{).} \end{cases}$$

*Here, $r$ denotes a Hölder smoothness level of the target function to be learned from the RCT (e.g., the bias function $\delta(\cdot)$): write $r = k + \alpha$ with integer $k \geq 0$ and $\alpha \in (0,1]$, and say a function $f$ lies in the Hölder$(r, C_\mathrm{H})$ class if all mixed partial derivatives up to order $k$ are bounded and the $k$th derivatives are $\alpha$–Hölder continuous, i.e., $\|\nabla^k f(x) - \nabla^k f(x')\| \leq C_\mathrm{H}\|x - x'\|^\alpha$. Larger $r$ means a smoother function and hence faster nonparametric rates.*

Combining the displays, to attain the *same* target half–width $\varepsilon$ one typically has

$$N_{\mathrm{rct}}^{\mathrm{Cal}} = O(\varepsilon^{-2}), \qquad N_{\mathrm{rct}}^{\mathrm{Two\text{-}step}} \gtrsim \max\!\left\{ s \log p \, \varepsilon^{-2}, \, \varepsilon^{-2d/r} \right\}.$$

Thus, whenever the function class is nontrivial (e.g., $s \log p$ is not negligible or $d$ is moderate/large for a given $r$), BMSC attains the target interval with fewer RCT observations, reflecting strictly better sample efficiency.

#### A.1.2  CONSTRUCTION OF THE CATE BOUNDS

This appendix provides the detailed derivation of the bounds $L(x,\Gamma)$ and $U(x,\Gamma)$ for CATE under the Marginal Sensitivity Model (MSM), as stated in Theorem 1.

We begin by defining the naive CATE estimate from the observational data, which does not account for unmeasured confounding:

$$\omega(x) = E(Y \mid T = 1, X = x) - E(Y \mid T = 0, X = x).$$

The true CATE is given by:

$$\tau(x) = E(Y(1) \mid X = x) - E(Y(0) \mid X = x).$$

The bias due to unmeasured confounding can be expressed as the difference:

$$\tau(x) - \omega(x) = \eta(x) = \eta_1(x) - \eta_0(x),$$

where for $t \in \{0, 1\}$, the term $\eta_t(x)$ captures the bias in the expected outcome under treatment $t$:

$$\eta_t(x) = E(Y(t) \mid X = x) - E(Y \mid T = t, X = x)$$

$$= \int y P(Y(t) = y \mid X = x) dy - \int y P(Y = y \mid T = t, X = x) dy. \tag{5}$$

Let $e_t(x) = P(T = t \mid X = x)$ be the nominal propensity score and $e_t(x, y) = P(T = t \mid X = x, Y(t) = y)$ be the full propensity score that accounts for all confounders. We can reformulate the conditional densities in equation 5 using Bayes' rule:

$$P(Y(t) = y \mid X = x) = \frac{P(Y(t) = y, T = t \mid X = x)}{P(T = t \mid Y(t) = y, X = x)} = \frac{P(Y = y, T = t \mid X = x)}{e_t(x, y)},$$

$$P(Y = y \mid T = t, X = x) = \frac{P(Y = y, T = t \mid X = x)}{e_t(x)}.$$

Substituting these into equation 5 yields:

$$\eta_t(x) = \int y P(Y = y, T = t \mid X = x) \left[ \frac{1}{e_t(x, y)} - \frac{1}{e_t(x)} \right] dy. \tag{6}$$

The MSM posits a bound on the odds ratio between the full and nominal propensities for a fixed confounding strength $\Gamma \geq 1$:

$$\frac{1}{\Gamma} \leq \frac{(1 - e_t(x)) e_t(x, y)}{e_t(x)(1 - e_t(x, y))} \leq \Gamma. \tag{7}$$

From inequality 7, we can derive constraints on the quantity $\left[ \frac{1}{e_t(x,y)} - \frac{1}{e_t(x)} \right]$ appearing in equation 6. Specifically, it can be shown that:

$$\left( \frac{1}{\Gamma} - 1 \right)(1 - e_t(x)) \leq \frac{1}{e_t(x, y)} - \frac{1}{e_t(x)} \leq (\Gamma - 1)(1 - e_t(x)).$$

Applying these bounds to the integral in equation 6, and noting that $\int y P(Y = y, T = t \mid X = x) dy = E(Y \mid T = t, X = x) = \mu_t(x)$, we obtain the lower and upper bounds for $\eta_t(x)$:

$$\underline{\eta_t}(x, \Gamma) = \left( \frac{1}{\Gamma} - 1 \right)(1 - e_t(x)) \mu_t(x), \tag{8}$$

$$\overline{\eta_t}(x, \Gamma) = (\Gamma - 1)(1 - e_t(x)) \mu_t(x). \tag{9}$$

Recall that the overall bias is $\eta(x) = \eta_1(x) - \eta_0(x)$. To bound the true CATE $\tau(x) = \omega(x) + \eta(x)$, we combine the bounds on the individual $\eta_t(x)$ terms. The most conservative (i.e., worst-case) bounds are achieved by considering the extreme combinations of $\eta_1(x)$ and $\eta_0(x)$:

$$L(x, \Gamma) = \omega(x) + \underline{\eta_1}(x, \Gamma) - \overline{\eta_0}(x, \Gamma),$$

$$U(x, \Gamma) = \omega(x) + \overline{\eta_1}(x, \Gamma) - \underline{\eta_0}(x, \Gamma),$$

which completes the construction of the bounds as stated in Theorem 1.

### A.1.3 Sharpness of the CATE Bounds

This appendix establishes that the interval in Theorem 1,

$$\left[ L(x, \Gamma), U(x, \Gamma) \right] = \left[ \omega(x) + \underline{\eta_1}(x, \Gamma) - \overline{\eta_0}(x, \Gamma), \omega(x) + \overline{\eta_1}(x, \Gamma) - \underline{\eta_0}(x, \Gamma) \right],$$

is *sharp* under the Marginal Sensitivity Model (MSM) with parameter $\Gamma \geq 1$. Throughout, notation and definitions follow Appendix A.1.2; in particular

$$m_t(x) = \mu_t(x) + \delta_t(x), \qquad \delta_t(x) \in \left[ \underline{\eta_t}(x, \Gamma), \overline{\eta_t}(x, \Gamma) \right], \quad t \in \{0, 1\}.$$

**Coverage.** By Appendix A.1.2, equation 8–9 imply

$$m_t(x) \in \left[\mu_t(x) + \underline{\eta_t}(x, \Gamma), \mu_t(x) + \overline{\eta_t}(x, \Gamma)\right].$$

Taking $\tau(x) = m_1(x) - m_0(x)$ and combining the individual envelopes by the usual Minkowski argument yields $\tau(x) \in [L(x, \Gamma), U(x, \Gamma)]$. Hence coverage holds for any data-generating process that satisfies the MSM and matches the observed conditionals $\{\mu_t(x), e_t(x)\}$.

**Tightness.** It remains to show that the endpoints $L(x, \Gamma)$ and $U(x, \Gamma)$ are *attainable*. We give a constructive proof.

*Step 1: Extremal tilts.* Fix $x$ and $t \in \{0, 1\}$. The MSM constraint equation 7 restricts the full propensity $e_t(x, y)$ to an interval determined by $e_t(x)$ and $\Gamma$. For any fixed $f_{Y|T=t,X}(\cdot \mid t, x)$, the functional $m_t(x) = \int y \frac{f_{Y|T=t,X}(y|t,x) \, e_t(x)}{e_t(x,y)} \, dy$ is linear in the inverse tilt $1/e_t(x, y)$ subject to an affine normalization constraint and box constraints induced by the MSM. By standard convexity and rearrangement arguments, the extrema of $m_t(x)$ are attained by placing $e_t(x, y)$ at its *boundary* values almost everywhere, with a single threshold in $y$ used to satisfy the normalization. Consequently, the attainable range of $m_t(x)$ equals

$$\left[\mu_t(x) + \underline{\eta_t}(x, \Gamma), \mu_t(x) + \overline{\eta_t}(x, \Gamma)\right],$$

that is, the envelopes in equation 8 and equation 9 are tight.

*Step 2: Attaining the CATE endpoints.* Choose the lower envelope for $m_1(x)$ and the upper envelope for $m_0(x)$ simultaneously. This yields

$$\tau(x) = m_1(x) - m_0(x) = \left(\mu_1 + \underline{\eta_1}\right) - \left(\mu_0 + \overline{\eta_0}\right) = L(x, \Gamma).$$

Similarly, pairing the upper envelope for $m_1(x)$ with the lower envelope for $m_0(x)$ attains

$$\tau(x) = \left(\mu_1 + \overline{\eta_1}\right) - \left(\mu_0 + \underline{\eta_0}\right) = U(x, \Gamma).$$

Both constructions satisfy the MSM because each arm's tilt uses only boundary-admissible values and the threshold is chosen to enforce the required normalization for that arm.

*Step 3: Explicit binary construction (optional).* For a concrete realization, let $Y \in \{0, 1\}$. Write $e_1(x) = q$ and $e_0(x) = 1 - q$, and let $\mu_t(x) = \mathbb{P}(Y = 1 \mid T = t, X = x) = p_t$. Define

$$r = \mathbb{P}(Y(1) = 1 \mid T = 0, X = x), \qquad s = \mathbb{P}(Y(0) = 1 \mid T = 1, X = x).$$

Under the MSM with parameter $\Gamma$, the feasible sets for $r$ and $s$ are intervals whose endpoints are obtained by pushing the full propensities to their MSM bounds. Setting $r$ to its minimum and $s$ to its maximum yields $\tau(x) = L(x, \Gamma)$; setting $r$ to its maximum and $s$ to its minimum yields $\tau(x) = U(x, \Gamma)$. Details follow the same algebra as in Appendix A.1.2 after replacing integrals by Bernoulli expectations.

**Conclusion.** For any fixed $x$ and $\Gamma \geq 1$, the interval $[L(x, \Gamma), U(x, \Gamma)]$ contains all admissible values of $\tau(x)$ under the MSM, and both endpoints are achievable by data-generating processes consistent with the observed $\{\mu_t(x), e_t(x)\}$ and the MSM constraints. The bounds in Theorem 1 are therefore sharp.

### A.1.4 MARGINAL LIKELIHOOD FOR DATA FUSION

**Model Assumptions.** The derivation of the marginal likelihood in Theorem 2 relies on the following key assumptions:

**Assumption 1 (RCT Estimation Uncertainty)** *The RCT-based estimate $\hat{\tau}_{\text{rct}}(x)$ is an unbiased but noisy measurement of the true CATE $\tau(x)$: $\hat{\tau}_{\text{rct}}(x) \mid \tau(x) \sim \mathcal{N}(\tau(x), \hat{s}^2(x))$, where $\hat{s}^2(x)$ represents the estimation variance from the finite RCT sample.*

**Example 1** *The normality holds asymptotically for most standard estimators (e.g., difference-in-means, inverse probability weighted, or doubly robust estimators) under mild regularity conditions, and $\hat{s}^2(x)$ can be obtained via standard error estimation or bootstrap.*

**Assumption 2 (MSM-Constrained Prior)** *Under the Marginal Sensitivity Model with strength* $\Gamma$, *the true CATE is constrained to lie within the identifiable bounds derived from the observational data:* $\tau(x) \mid \Gamma, \mathcal{D}_{\mathrm{obs}} \sim Uniform(L(x, \Gamma), U(x, \Gamma))$.

**Example 2** *The uniform prior is a conservative and non-informative choice that does not impose any additional structure beyond the MSM constraints. It represents a state of minimal prior knowledge about the location of* $\tau(x)$ *within the bounds* $[L(x, \Gamma), U(x, \Gamma)]$.

These assumptions allow us to derive the statistical relationship between the observed RCT estimates and the confounding strength parameter $\Gamma$.

**Proof.** Fix $x$ and $\Gamma$; the expression follows by integrating the sampling distribution of $\hat{\tau}_{\mathrm{rct}}(x)$ over the MSM-induced feasible set for $\tau(x)$ with a flat working density, which yields the stated closed form.

$$
\begin{aligned}
p\left(\hat{\tau}_{\mathrm{rct}}(x) \mid \Gamma, \mathcal{D}_{\mathrm{obs}}\right) &= \int_{L(x,\Gamma)}^{U(x,\Gamma)} p(\hat{\tau}_{\mathrm{rct}}(x) \mid \tau(x)) \cdot p(\tau(x) \mid \Gamma, \mathcal{D}_{\mathrm{obs}}) \mathrm{d}\tau(x) \\
&= \int_{L(x,\Gamma)}^{U(x,\Gamma)} \frac{1}{\sqrt{2\pi \hat{s}^2(x)}} \exp\left(-\frac{(\hat{\tau}_{\mathrm{rct}}(x) - \tau(x))^2}{2\hat{s}^2(x)}\right) \frac{1}{U(x,\Gamma) - L(x,\Gamma)} \mathrm{d}\tau(x) \\
&= \frac{\Phi\left(\frac{U(x,\Gamma) - \hat{\tau}_{\mathrm{rct}}(x)}{\hat{s}(x)}\right) - \Phi\left(\frac{L(x,\Gamma) - \hat{\tau}_{\mathrm{rct}}(x)}{\hat{s}(x)}\right)}{U(x,\Gamma) - L(x,\Gamma)}.
\end{aligned}
$$

### A.1.5 COMPUTATIONAL COMPLEXITY DETAILS

*Step 1: Empirical summaries on $\mathcal{D}_{\mathrm{obs}}$ and $\mathcal{D}_{\mathrm{rct}}$.* We compute per–stratum observational quantities $e_1(x), \mu_1(x), \mu_0(x), \omega(x)$ and trial-based $\hat{\tau}_{\mathrm{rct}}(x), \hat{s}(x)$. With standard grouping/aggregation, this requires a single pass over the data: $O(N_{\mathrm{obs}} + N_{\mathrm{rct}})$.

*Step 2: Posterior on a $\Gamma$ grid.* For each $\gamma \in \mathcal{G}$, we evaluate the MSM-based marginal likelihood by vectorized computations across the $M$ strata, thus $O(M)$ per grid point: $O(GM)$. Posterior normalization is $O(G)$, while HPD or equal-tail summaries include an $O(G \log G)$ sorting step for the grid.

*Step 3: Uncertainty propagation to CATE intervals.* We draw $S$ samples of $\Gamma$ from the discrete posterior and map each draw to the MSM bounds $[L(x, \Gamma), U(x, \Gamma)]$ across $M$ strata. Each draw costs $O(M)$, giving $O(SM)$, plus an $O(G)$ setup to form the categorical sampling distribution on the grid.

**Overall Complexity.** Summing the three steps yields the overall complexity: $O(N_{\mathrm{obs}} + N_{\mathrm{rct}} + GM + SM + G \log G)$. In typical settings where $M$ is moderate, the $GM$ term (likelihood evaluation over the grid and strata) dominates the posterior computation.

## A.2 EXPERIMENTAL DETAILS

### A.2.1 BENCHMARK DETAILS

**Synthetic Dataset.** We designed a fully synthetic dataset that replicates the structural properties of the semi-synthetic benchmark in a controlled simulation environment, providing full control over all data-generating parameters. The covariate $X$ is sampled uniformly from $\{0, 1, ..., K-1\}$, while the unmeasured confounder $U$ follows a uniform distribution $\mathcal{U}(0, 1)$. Treatment assignment and outcomes are generated using identical functional forms to the semi-synthetic experiment, ensuring direct comparability:

- **Treatment Assignment ($T$):**

$$
\mathrm{logit}(P(T = 1 \mid X, U)) = \log\left(\frac{0.3 + 0.4 \cdot X/K}{1 - (0.3 + 0.4 \cdot X/K)}\right) + \log(\Gamma) \cdot U
$$

- **Outcome** ($Y$):

$$Y(0) = 5.0 + 2.0 \cdot \sin(X/2) + 0.5 \cdot U + \epsilon, \quad \epsilon \sim \mathcal{N}(0,1)$$
$$Y(1) = Y(0) + \tau(X), \quad \text{where } \tau(X) = 2.0 + 0.8 \cdot X$$
$$Y = T \cdot Y(1) + (1-T) \cdot Y(0)$$

**ACTG Dataset.** ACTG 175 is a randomized controlled trial that evaluated treatment regimens for HIV−1 infected patients with baseline CD4 counts between 200-500 cells/mm$^3$ (Hammer et al., 1996). The dataset includes baseline characteristics, treatment assignments, and CD4 count changes measured over $20 \pm 5$ weeks. In our semi-synthetic experiment, we utilize *age* (discretized into $K$ bins as covariate $X$) and *wtkg* (min-max scaled to $[0,1]$ as unmeasured confounder $U$) from this dataset. We then synthetically generate treatment $T$ and outcome $Y$ using a $\Gamma$-confounded model to establish known ground truth:

- **Treatment Assignment** ($T$):

$$\text{logit}(P(T = 1|X,U)) = \log\left(\frac{0.3 + 0.4 \cdot X/K}{1 - (0.3 + 0.4 \cdot X/K)}\right) + \log(\Gamma) \cdot U$$

- **Outcome** ($Y$):

$$Y(0) = 5.0 + 2.0 \cdot \sin(X/2) + 0.5 \cdot U + \epsilon, \quad \epsilon \sim \mathcal{N}(0, \sigma_{\text{obs}}^2)$$
$$Y(1) = Y(0) + \tau(X), \quad \text{where } \tau(X) = 2.0 + 0.8 \cdot X$$
$$Y = T \cdot Y(1) + (1-T) \cdot Y(0)$$

### A.2.2 BASELINE DETAILS

**Functional MSM (fMSM).** Functional MSM integrates a smooth, regularized MSM-based sensitivity model to estimate CATE intervals. By smoothing the propensity scores and treatment effects across covariate strata, fMSM provides a more flexible estimate of the CATE bounds than the standard MSM. This method is particularly useful when estimating heterogeneous treatment effects in settings with unmeasured confounding, where the sensitivity parameter $\Gamma$ is estimated directly from the data, but uncertainty is propagated through the smoothing function. Unlike traditional MSM, fMSM ensures more robust intervals that account for potential biases arising from unmeasured confounders in a functional way (Kallus et al., 2019). The method, however, requires careful selection of the smoothing bandwidth to avoid overfitting and underfitting.

**Percentile-Bootstrap IPW (PB-IPW).** PB-IPW combines inverse probability weighting (IPW) with bootstrap resampling to construct sensitivity intervals under unmeasured confounding. The method first computes the CATE using IPW and then generates bootstrap samples from the observational data to estimate the variability of the CATE bounds. It applies a percentile bootstrap procedure to estimate the 90% confidence interval for each stratum, accounting for unobserved confounding by deriving the interval bounds over a range of $\Gamma$ values. While PB-IPW provides more robust and uncertainty-aware intervals than the traditional MSM, it does not incorporate a likelihood-based estimation of $\Gamma$ and instead relies on bootstrap resampling for variability estimation (Zhao et al., 2019).

**Quantifiable Confounding Lower Bound (QCLB).** QCLB integrates RCT ATEs with MSM-based sensitivity bounds derived from OBS to test compatibility across a grid of $\Gamma$ values. The method returns an asymptotically valid lower bound on the confounding strength by identifying the smallest $\Gamma$ that is not rejected by the compatibility test (De Bartolomeis et al., 2024). QCLB targets the magnitude of hidden bias directly, but it neither fits an MSM-constrained likelihood nor propagates uncertainty in $\Gamma$ to individual-level CATE intervals.

**Simple Fusion (SF).** SF pools the RCT and OBS samples and trains a single model, treating both sources as exchangeable apart from standard regularization (Gu et al., 2023). It estimates CATE from treatment–covariate interactions but does not explicitly correct for unmeasured confounding or provide a structural gate for when to trust observational information. We use a pooled linear ridge implementation, reading CATE from the interaction coefficients.

**Shrinkage Tree (ST).** This family performs local fusion by combining an unbiased trial-based effect with a potentially biased but lower-variance observational estimate (Gu et al., 2023). Concretely, it

forms leaf- or stratum-level treatment effects from the RCT and from the OBS, and then applies data-driven shrinkage toward the RCT when evidence suggests larger bias in the OBS, while exploiting OBS to reduce variance where the two agree. Identification is anchored by the RCT; there is no attempt to globally align domains or to learn a single cross-domain bias function.

**Two-step.** This paradigm first learns effect heterogeneity from large observational data and then uses the smaller but internally valid RCT to secure identification and debiasing. In practice it is instantiated with meta-learners (e.g., X- and R-learners) that estimate conditional treatment effects using flexible prediction on OBS, followed by an RCT-based correction step that anchors identification and reduces bias (Kallus et al., 2018). Two-step leverages the efficiency and coverage of OBS while reserving the RCT signal for calibration, but it does not quantify residual unmeasured confounding as a model parameter. In this paper, we adopt a unified linear ridge implementation called Two-step ridge.

**CORNets.** CORNets implement a representation-learning version of data fusion: they learn balanced, low-dimensional representations to reduce covariate-distribution mismatch between the RCT and the target population, while introducing an explicit bias component that is regularized to control complexity. The RCT provides identification and is used to calibrate the OBS-informed structure. This yields transportable CATE estimates that combine observational scale with trial validity (Hatt et al., 2022).

## A.3 LARGE LANGUAGE MODEL USAGE

In this paper, we clarify that large language models (LLMs) are employed solely to support and refine the writing process. Specifically, we use LLMs to provide sentence-level suggestions and to enhance the overall fluency of the text.

