# OpenReview forum: "Adaptive Marginal Sensitivity with Limited RCT Data for CATE Estimation"
_ICLR.cc/2026/Conference — Submitted to ICLR 2026_

### Official Review · Reviewer_YPDe · 2025-10-17

**Soundness:** 1
**Presentation:** 3
**Contribution:** 1
**Rating:** 2
**Confidence:** 4

**Summary:**

The paper proposes a Bayesian method for data fusion when both randomized and observational data are available. It leverages the marginal sensitivity model (MSM) to infer partial identification bounds of the CATE on the observational data. The sensitivity parameter itself is estimated in a Bayesian manner via the experimental data.

**Strengths:**

- Inferring the strength of unobserved confounding via data fusion is an interesting and relevant problem
- The paper is written intuitively and is easy to follow

**Weaknesses:**

- The assumptions required for the proposed approach are not entirely clear. Is the transportability of the CATE required (i.e., assuming the same CATE across observational and experimental data)? That would be a strong assumption often violated in practice, but I don't see how the proposed method would work otherwise.
- The aim of the paper is not entirely clear. Is the goal to infer gamma to quantify confounding strength within the observational data, or is it to infer the CATE? I doubt the latter would benefit from inferring gamma but would rather require a data fusion approach, as in, for example, "Removing Hidden Confounding by Experimental Grounding" by Kallus et al. Additionally, I am not convinced why we would ever need to report the partially identified CATE interval for RCTs. EIther, (i) we should use the identified RCT estimator and report confidence intervals, or (ii) report an observational-data enhanced estimator (e.g., from Kallus) and also obtain confidence intervals. This way we obtain an asymptotically consistent estimator with vanishing interval size, while the partial identification bounds can be non-informative and do not get tighter with increasing sample size.
- Related to the previous point, it is unclear to me why the proposed method outperforms data-fusion methods. Estimating gamma should not be necessary when the target is CATE estimation; a simple debiasing approach should be sufficient. I suspect that the improvements are due to modeling choices and not due to the actual benefit of estimating gamma.
- Unclear theoretical claims. Theorem 1 and its proof are known in the literature. I am not sure if the authors are claiming novelty here, but if this is simply copied, credit should be given to the corresponding sources.
- Marginal contribution & novelty. The idea of inferring gamma from experimental data is not novel. To my understanding, the contribution is the Bayesian approach, which is somewhat straightforward.

**Questions:**

- What is the intuition behind the proposed method outperforming classical data fusion methods? Why would we ever need gamma for this task?
- What is the motivation for reporting partial identification bounds on identified RCTs? Admittedly, RCTs yield higher variance estimators due to limited sample size, but this should be acknowledged via (possibly observationally-enhanced) confidence intervals and not partial identification bounds.
- How are model architectures/ hyperparameters chosen for the method and baselines?

---

### Official Review · Reviewer_xQxK · 2025-10-31

**Soundness:** 3
**Presentation:** 2
**Contribution:** 3
**Rating:** 6
**Confidence:** 3

**Summary:**

This paper proposes a data fusion method for conditional average treatment effect estimation with unobserved confounders. Based on the marginal sensitivity model, the submission proposes a Bayesian method to estimate the sensitivity parameter $\Gamma$ from data. Theoretical analysis and empirical results are provided to evaluate the performance of the proposed method.

**Strengths:**

1. CATE estimation with unobserved confounders is an important and interesting problem.

2. The proposed method is generally reasonable.

**Weaknesses:**

1. My major concern is about the prior of the sensitivity parameter $\Gamma$. Although the method does not need to heuristically set a value for $\Gamma$, it is still necessary to determine the prior $\pi(\Gamma)$ and the range of $\Gamma$, which plays a similar role to the setting of the value of $\Gamma$. In other words, the subjective specified $\Gamma$ is another kind of prior information. What is the difference between these two kinds of prior information?

2. The estimation error of $\Gamma$ could affect the results of CATE. How to measure and reduce such an effect?

3. It could be interesting to evaluate the performance of $\Gamma$ estimation with different numbers of RCT samples. Although Table 2 conducts a simple experiment, it is difficult to draw meaningful conclusions based on only two values of $N_{rct}$.

**Questions:**

Please refer to the weaknesses.

---

### Official Review · Reviewer_6nLz · 2025-11-01

**Soundness:** 3
**Presentation:** 3
**Contribution:** 3
**Rating:** 6
**Confidence:** 2

**Summary:**

This paper proposes a new framework, Bayesian Marginal Sensitivity Calibration (BMSC), to address a critical limitation in MSM: the subjective specification of the sensitivity parameter $\Gamma$. BMSC learns its posterior distribution by OBS+RCT.

The method first uses OBS data to construct a $\Gamma$-dependent CATE interval. It then treats the RCT's CATE as a noisy, unbiased measurement of the true $\tau(x)$ and derives a closed-form marginal likelihood combined with a prior $\pi(\Gamma)$ to yield a posterior distribution for the confounding strength, $P(\Gamma | \mathcal{D}_{rct}, \mathcal{D}_{obs})$. This yields a data-driven estimation instead of subjective specification.

**Strengths:**

1. The paper addresses a key practical barrier in sensitivity analysis—the subjective choice of $\Gamma$.
2. The framework is technically sound. However, I am not an expert in SMS analysis, so I don't check whether the theorems and proofs (Thms 1-3) are rigrous.
3. The paper is well-written. The figures and algorithms clearly illustrate the method's workflow and contrast it with prior work.

**Weaknesses:**

1. Strong Assumption: I am not sure whether Assumption 2 is too strong or if it is a common assumption when analyzing MSM. Could you please provide more explanations to justify whether the assumption is realistic?

2. The Prior $\pi(\Gamma)$: The method replaces the subjective choice of a single $\Gamma$ with the subjective choice of a prior $\pi(\Gamma)$. The author also confirms that prior dispersion in the confounding strength affects accuracy. Could you please provide more information to assist us in selecting prior $\pi(\Gamma)$?

**Questions:**

See Weaknesses.

---

### Official Review · Reviewer_wsay · 2025-11-01

**Soundness:** 2
**Presentation:** 3
**Contribution:** 2
**Rating:** 2
**Confidence:** 3

**Summary:**

The paper proposes Bayesian Marginal Sensitivity Calibration (BMSC), a pipeline which allows to learn the sensitivity parameter $\Gamma$ used to assess the confounding strength in observational trials, thus allowing for CATE estimation without the need to pre-specify $\Gamma$. The work combines the frameworks of Marginal Sensitivity Models (MSMs) and data fusion (combining data from a (small) RCT and an observational trial). Using a specified prior for $\Gamma$ and a grid over this prior, the algorithm computes the MSM lower and upper bounds and the marginal likelihood for each $\gamma$, which is used to compute the final posterior for $\Gamma$. Finally, the posterior mean or median is used for CATE interval estimation. The findings are illustrated with synthetic and semi-synthetic experiments.

**Strengths:**

The paper is well-structured overall and addresses a very relevant problem of estimating CATE from large-scale but potentially confounded observational trials. The objective of reducing the number of samples in the RCT required for data fusion-based CATE interval estimation is especially important, since RCT data is much more costly than observational data, as highlighted by the authors. Additionally, the experimental results in Section 5 look promising, and the results for smaller RCT sample sizes seem to be an improvement over the baselines.

**Weaknesses:**

1. The separation between novel contributions and existing results (e.g. Theorem 1) is unclear, and the first four pages feel redundant. Several statements are labeled as "Remarks" (e.g. Remark 1, Remark 2) that would read better as standard paragraphs. Assumptions for key theorems (e.g. Theorem 2) are missing or only partially stated in the main text; this obscures the scope and impact of the contribution and undermines confidence in the theory.
2. Novelty: The work appears to assemble established components (MSM envelopes, simple grid posterior) with the main novelty being the posterior construction for $\Gamma$. Several proofs read as straightforward applications of Bayes’ rule rather than introducing fundamentally new ideas.
3. Assumptions behind Theorem 2. Assumption 1 (Appendix A.1.4) models the RCT-based CATE estimate as $\hat{\tau}(x) = \tau(x) + \mathcal{N}(0,\hat{s}^2(x))$, where $\hat{s}^2(x)$ is a random quantity depending on the RCT. At minimum, the variance term should be treated as a (conditionally) non-random population quantity. In any case, the conditional Gaussianity is restrictive and should be discussed in the main text. Additionally, this requires an analysis of how the method behaves (for small sample sizes) if this assumption is violated.
4. The estimator is not purely data-driven, and for good performance, it requires 1) choosing a good prior for $\Gamma$ (especially $\Gamma_\max$); 2) the noise model for the CATE being close to Gaussian. Ablations from the authors on how robust their method is to misspecifications of these parameters would help strengthen the paper.
5. The authors repeatedly claim to conduct real-data experiments. However, the discussed experiments are semi-synthetic. This should be stated clearly from the beginning, and there should be a clear discussion of the design choices for the semi-synthetic experiments. Additionally, the paper would profit from a full real-data experiment.
6. If I am not mistaken, the $\Gamma$ posterior is used for computing its mean/median, which is then used as a plug-in estimator to compute the CATE intervals (Algorithm 1). However, the authors also claim that their method propagates the uncertainty about $\Gamma$ as a novel contribution. I don't think plugging in the mean propagates uncertainty though, as a delta distribution around $\mu$ and a wide Gaussian around $\mu$ have the same mean and median. Is something different to Algorithm 1 implemented in the experiments?
7. Sample-efficiency analysis: Appendix A.1.1 is imprecise: the constant $\kappa$ is not specified, and the error from the first-order delta method is not analyzed. The comparison to functional-estimation rates is also uneven: if one truly propagates the full posterior over
$\Gamma$, the effective rates and constants may differ. A careful finite-sample analysis (with explicit constants) is needed to substantiate claims of improved efficiency for small RCT sizes.
8. In the empirical evaluation, confidence intervals (or at least standard errors) are not reported, even though all experiments are controlled,  this would be straightforward and informative.
9. There seems to be no coverage analysis for the semi-synthetic experiment.
10. The manuscript contains multiple typos throughout the text, e.g. "to migrate this" instead of "to mitigate this".

**Questions:**

1. Could the authors clarify what they mean by "propagation of uncertainty" in terms of $\Gamma$ estimation?
2. Could the authors elaborate on Assumption 2, and illustrate whether the conditional Gaussianity assumption is violated in their experiments? How sensitive are the results to estimation of $\hat{s}$?
3. How do the results depend on the choice of $\Gamma_\max$ and the choice of the grid?
4. QCLB aims to provide a lower bound for $\Gamma$, could you elaborate how exactly you utilized QCLB for CATE MAE computation?
5. Is there a reason QCLB was omitted as a baseline in Figure 3?
6. Could you provide confidence intervals for the controlled experiments averaged over multiple simulations?

---

### Official Review · Reviewer_nnee · 2025-11-02

**Soundness:** 3
**Presentation:** 3
**Contribution:** 2
**Rating:** 4
**Confidence:** 3

**Summary:**

This paper addresses the problem of estimating CATE from observational data in the presence of unmeasured confounding. It targets the key limitation of MSM: the sensitivity parameter $\Gamma$ must be subjectively specified, and existing methods to calibrate $\Gamma$ using RCT data require impractically large samples. The method is a framework named Bayesian Marginal Sensitivity Calibration (BMSC). The idea behind the method is to learn the sensitivity parameter $\Gamma$ by fusing observational data with limited RCT data. It constructs $\Gamma$-dependent CATE bounds from the observational data, then uses a Bayesian model to find the posterior for $\Gamma$ that best aligns these bounds with the unbiased estimates from the RCT. They demonstrate the method using synthetic and semi-synthetic (ACTG) datasets, comparing BMSC against traditional MSM approaches and other data-fusion baselines. The findings are that BMSC effectively estimates $\Gamma$ and produces valid, sharp CATE intervals even when the RCT sample size is very small, demonstrating superior sample efficiency and lower estimation error than baseline methods in these scarce-data scenarios.

**Strengths:**

The paper's primary strength is its clarity. It is very well-written, well-organized, and easy to follow. The problem of subjective $\Gamma$ specification is well-motivated and explained; the BMSC method is presented logically, and the illustrations and experimental results are clear and well-presented.
While the general concept of using RCT data to calibrate sensitivity parameters is a known line of inquiry, the paper's main contribution is in providing a self-contained, practical, and well-executed Bayesian framework for this task.

**Weaknesses:**

The paper's primary weakness lies in whether its proposed solution is truly an improvement over the problem it aims to solve. The motivating problem is that subjectively specifying the sensitivity parameter $\Gamma$ is difficult and undermines objectivity. However, the BMSC framework does not fully eliminate this subjectivity; it merely trades one difficult assumption for a new set of strong, and arguably just as problematic, assumptions about the data-generating process.

Specifically, the method relies on several strong assumptions:It assumes the RCT-based CATE estimate, $\hat{\tau}_{rct}(x)$, follows a Normal distribution (Assumption 1). This is an asymptotic approximation. This assumption is directly at odds with the paper's central claim of working well in the "limited RCT sample" scenarios, where such an approximation is least likely to hold.

It also assumes a Uniform prior for the true CATE $\tau(x)$ within the observational bounds $[L(x,\Gamma), U(x,\Gamma)]$ (Assumption 2). This is a very strong assumption of "non-informativeness" that is not justified and may not be realistic; if confounding biases the true CATE toward one of the bounds, this assumption would be violated.


The method does not fully escape subjective specification. The paper's own sensitivity analysis (Section 5.4) clearly shows that the estimation of $\Gamma$ is sensitive to the choice of the prior $\pi(\Gamma)$. Critically, it shows that "more diffuse priors degrade accuracy." This means a practitioner must still make a subjective choice about the prior, and the most "objective" or "non-informative" choice (a diffuse prior) apparently leads to worse results.

**Questions:**

1. How do the authors justify that this new set of assumptions is practically better or less subjective than the original problem of just specifying $\Gamma$?

2. Could the authors clarify the primary advantage of BMSC over the QCLB baseline? Both methods use RCT data for calibration. Is the superior performance of BMSC in the low-sample setting (Table 2) primarily due to the move from a frequentist lower bound to a full Bayesian posterior? Or is this gain, as suggested by the sensitivity analysis, heavily driven by the choice of prior $\pi(\Gamma)$? How can the authors demonstrate that this advantage is not an artifact of the prior aligning well with the synthetic experimental setup?

3. What happens if assumption 2 is violated?

---

### Meta-Review · Area_Chair_5UCq · 2025-12-25

**Summary:**

The paper proposes a bayesian method to learn the senstivity parameter $\Tau$ in a marginal sensitivity model often used in CATE estimation under unobserved confounding by combining observational and RCT data. While RCT data (large sample) is typically used to calibrate $\Tau$, the claim is that often a large sample is not available, making it challenging to set a reasonable $\Tau$. Therefore the authors propose to impose a prior over $\Tau$ and learn a Bayesian posterior.

1. Reviewers mention that choice of $\Tau$ is replaced with the choice of choosing a good prior, which is challenging.
2. The work proceeds by making assumptions of normality on CATE, which is unrealistic.
3. The paper also assumes uniform distribution over bounds on the CATE, which is unrealistic and makes the method fragile. In terms of the writing, the distinction between the contributions and prior work needs to be more precise.
4. Reviewers had several clarifying question about the method itself, such as the fact that the mean/median of $\Tau$ are used as plug-in estimates over the CATE which does not quite propagate uncertainty, the normality assumption, external validity/transportability between RCT and observational data etc.

**Reviewer Concerns:**

There was no rebuttal and I do not think reviewer concerns were addressed.

**Reviewer Scores:**

No change

---

### Decision · Program_Chairs · 2026-01-26

Reject